# Land Use Changes in the Southeastern United States: Quantitative Changes, Drivers, and Expected Environmental Impacts

**Ryan Nedd [1] and Aavudai Anandhi [2],***

[1] School of the Environment, Florida Agricultural and Mechanical University, 1515 S. MLK Jr. BLVD, Tallahassee, FL 32307, USA

[2] Department of Biological Systems Engineering, Florida Agricultural and Mechanical University, Benjamin Banneker-Tech C, 1409 Wahnish Way, Tallahassee, FL 32307, USA

* Correspondence: anandhi.swamy@famu.edu

**Abstract:** Land use change analysis provides valuable information for landscape monitoring, managing, and prioritizing large area conservation practices. There has been significant interest in the southeastern United States (SEUS) due to substantial land change from various economic activities since the 1940s. This study uses quantitative data from the Economic Research Service (ERS) for landscape change analysis, addressing land change among five major land types for twelve states in the SEUS from 1945 to 2012. The study also conducted a literature review using the PSALSAR framework to identify significant drivers related to land type changes from research articles within the region. The analysis showed how each land type changed over the period for each state in the time period and the percentage change for the primary drivers related to land use change. The literature review identified significant drivers of land use and land cover change (LULCC) within the SEUS. The associated drivers were categorized into natural and artificial drivers, then further subdivided into eight categories related to land type changes in the region. A schematic diagram was developed to show land type changes that impacted environmental changes from various studies in the SEUS. The results concluded that Forest land accounted for 12% change and agricultural land for 20%; population growth in the region is an average of 2.59% annually. It also concluded that the need for research to understand past land use trends, direction and magnitude of land cover changes is essential. Significant drivers such as urban expansion and agriculture are critical to the impending use of land in the region; their impacts are attributed to environmental changes in the region and must be monitored.

**Keywords:** Southern United States; land use change; analysis; drivers; quantitative changes

## 1. Introduction

The Southeastern United States (SEUS) has undergone various land use changes that have led to environmental changes [1,2]. This large landscape has seen significant growth and changes over the century. Some examples include urban development [3], forest land clearing [4], agricultural expansion [5], climate variability [6], hydroclimate changes [7], temperature variability [8], and hydrological variations [9]. The region's land change has detected trends leading to land degradation at an estimated rate of 5.8% since 1973 [10]. Some diverse drivers result in extensive landscape changes; however, the major drivers (e.g., natural disasters, urbanization, agriculture, forestry) observed has caused the most effects [11]. Increasing importance was given to the essential drivers as a result of land use and land cover change (LULCC). While these drivers are natural and artificial, anthropogenic LULCC have been identified with significant climate change [12]. One of the primary drivers of LULCC is agriculture. According to the Economic Research Service (ERS) 2012 report, cropland in the SEUS account for approximately 14.4% (90.3 million

acres) of cropland [13], and the SEUS accounts for 17% of agricultural production in the USA [14,15]. Its association is due to its current large-yielding output of landscape changes in the region [4,16]. Another significant driver identified is urbanization [17]. The effects have affected the region's climate [3], hydrology [18], ecosystem services [19,20], carbon sequestration by forest [21], and policies relating to land use [22]. These driving forces of LULCC have influenced environmental changes in the region, leading to a shift in landscape use. Through land-use classification with data from the National Land Cover Database (NLCD), researchers have identified various land types in the USA with remote sensing [23]. Multiple federal agencies use these land types for land monitoring and management through multi-temporal remote sensing for various land types [24]. The ERS is a federal agency that compiles datasets and reports using LULCC estimates in the USA. These land types are identified separately, reported for each state, and used by researchers to determine future land use based on land use trends [25].

Land use analysis is an essential part of understanding landscape dynamics in the SEUS [26]. This analysis helps researchers organize, interpret and report on the findings of these analyses [27]. Advanced maps, multiple datasets, and models have been used for monitoring and decision-making for land use analysis [28]. The identification of changing land types over time, the various drivers that lead to land changes, and the quantitative values shown from these changes is a synopsis of land use analysis [29]. Various socioeconomic, technical, and ecological differences between the states make up the SEUS; however, most of their challenges remain the same when analyzed [30]. As a result of changing landscapes in the SEUS, an analysis of land resources provided is suitable for evaluating changing landscapes over time [31]. Multiple datasets have been compiled on land use information by federal agencies in the USA. The National Agricultural Statistical Service (NASS), along with the USDA Forest Service and Farm Service Agency (FSA), are agencies that provide data/information for agricultural land use and forestry for analyzing information for particular areas in land use [32]. This information is made public for scientists and researchers to use for land use analysis, which provides a foundation for the impacts of LULCC.

While land use assessments have been carried out in various forms globally [28]. The general goal of this study is to analyze quantitative land cover data using a simple technique and review the literature on the drivers that lead to LULCC in the SEUS. This came as a recommendation to understand LULCC over various periods and their effects on future landscapes [33]. Some researchers would have estimated these changes in the entire United States [34,35] and others in specific states [36]. Earlier studies have analyzed these changes in the region using various methods, such as remote sensing and modeling spatial and temporal data [3,37–39]. Some methods of analysis involve change detection techniques (e.g., Ratioing, principal component analysis, and change vector analysis) of various datasets, maps, and satellite images to determine land use change [40]. Our goal is to use a simple algebraic analysis technique known as "differencing" to show changes in land type over a specific period. This simple method works well with acreage data and can show the differences between time frames. The major land types from the ERS 2012 report within the United States Department of Agriculture (USDA) were eleven land types [13]. This research will group these land types into five major categories, these include (1) cropland total, (2) forest land total; (3) grassland total; (4) special land use area total; (5) urban land total. A specific framework protocol for collecting and analyzing published research articles will be used to perform the literature review. This information will be used to give results related to the drivers related to landscape changes in the SEUS. The specific objectives are 1. To estimate LULCC from observed data for five (5) major land types during 1945–2012 in twelve states in the SEUS. 2. To synthesize the drivers and expected environmental impacts from literature in the SEUS. This information will give a better understanding of the past land use trends in the region and their impacts on future land use. The overall aim of this study will enhance existing scientific knowledge related to LULCC in the SEUS.

## 2. Materials and Methods

### 2.1. Study Area

The region examined for this analysis is the SEUS. It is divided into physiographic areas: Appalachians, Piedmont, and Coastal Plain [41]. These ecoregions include the states of Alabama, Arkansas, Florida, Georgia, Kentucky, Louisiana, Mississippi, North Carolina, South Carolina, Tennessee, Virginia, and West Virginia. These states occupy the southeastern portion of the United States (Figure 1) and are characterized by a mild, wet, sub-tropical climate, with an average annual temperature of 17 °C and annual precipitation greater than 750–1300 mm that is rapidly changing as a result of LULCC [5,42,43].

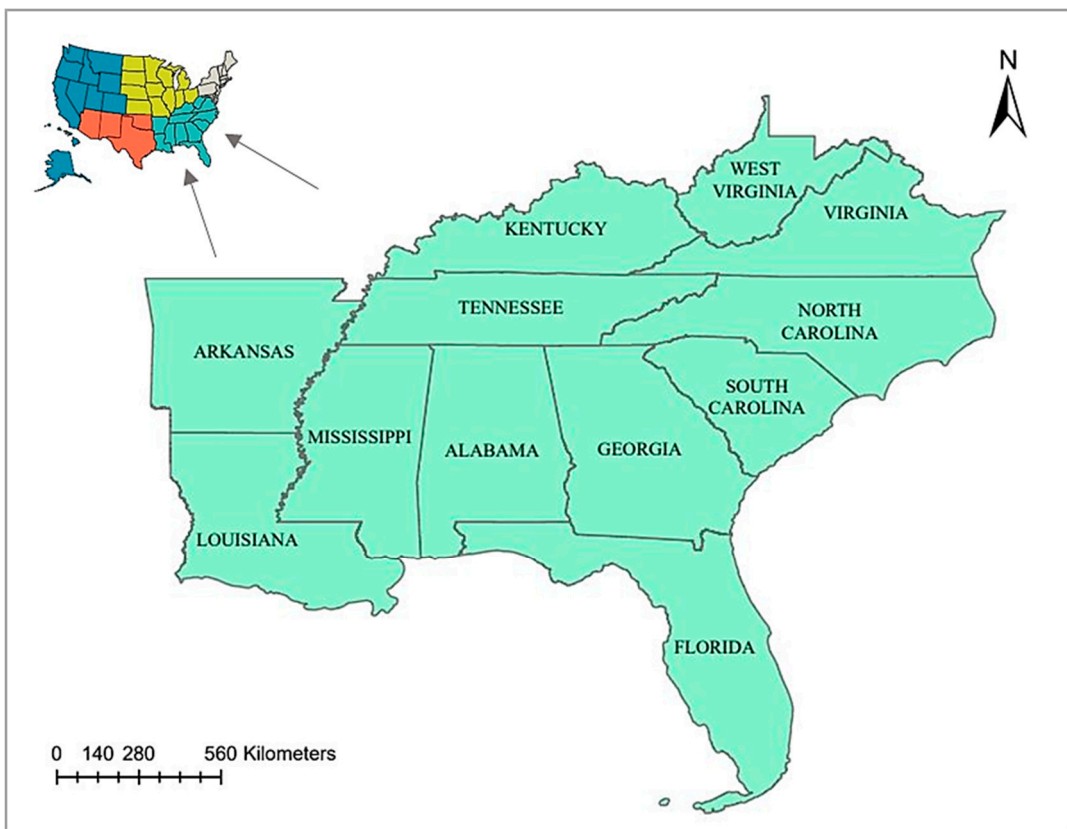

**Figure 1.** Showing the southeastern region of the United States of America.

This region is also characterized by multiple river basins and hydrological processes [9,18], rapid land use and land cover changes [3,44,45], a wide wildlife species distribution [46], and various forest types [45]. The climate has favored, over time, intense agricultural exploitation, intense timber exploitation, strong economic growth, population, and urbanization [3,15]. Agriculture and urbanization continue to be paramount in this region, with an estimated 77 million persons living in the region [19]; agriculture has estimated to be about $56 billion annually [15]. This region continues to undergo numerous changes because of LULCC.

### 2.2. Synthesis of LULCC, Drivers, and Environmental Impacts from Published Articles in the SEUS

This research used a Literature Review to collect journal articles for analysis. The PSALSAR framework was developed by Mengist et al. [47] and selected as the literature review tool for this research. It uses a process of Protocol (P) to set research questions; Search (S) all relatable studies to the research topic; Appraisal (A) of relevant articles according to the research criteria; Synthesis (S) of articles; Analysis (A) of articles for quality and efficacy; and Report (R) on the research findings in a scientific format [47]. This framework used research articles obtained through the search engine Google Scholar

(https://scholar.google.com/ accessed on 8 December 2022) and collected from 2000 to 2022 for assessment according to the framework's protocol.

This research downloaded and compiled articles based on keywords for research objective two. The search keywords used are as follows: (1) Land use and land cover change; (2) Southeastern, United States; (3) Drivers of Land use and land cover change. This literature review summarized the information from the research articles for objective two in the SEUS. The findings were then presented in two illustrations to show drivers and environmental changes.

*2.3. The Steps Used According to the PSALSAR Framework for Collecting and Synthesizing Articles*

The scope and terms "land use and land cover change;" "Southeastern, United States;" and "Drivers of Land use and land cover change" was used as the keywords for the search strategy.

Step 1. A total of twenty (20) articles on "Drivers of Land use and land cover change in the Southeastern United States" were downloaded according to the PSALSAR framework.

Step 2. All articles were used to develop an illustration to show the natural and artificial drivers of LULCC in the SEUS.

Step 3. Articles were used to develop an illustration to show LULCC leading to environmental change in the SEUS.

## 3. Analysis of Major Land Types Using Observed Quantitative Data

In the observed data, this research uses quantitative data and an algebraic-based technique (Differencing) for analyzing the various land use type totals for objective 1. The datasets used for this analysis were compiled from the Economic Research Service (ERS), United States Department of Agriculture (USDA), as shown in Table 1. The "Difference" of acreage data between time periods was calculated and used to show land type changes. The federal data used in this research was acquired from the ERS website for general land use and land cover estimates within the U.S. https://www.ers.usda.gov/about-ers/partnerships/strengthening-statistics-through-the-icars/land-use-and-land-cover-estimates-for-the-united-states/ (accessed on 8 December 2022).

**Table 1.** Showing the definitions and descriptions of various land types (Source: Bigelow and Borchers, [13]).

| Land Type | Categories | Explanation |
|---|---|---|
| Forest land | Definition | The forest use excludes forest land in parks, wildlife areas, and similar special-purpose uses from the U.S. Forest Service's inventory of total forestland. |
| | Forest land grazed | It mainly includes forest acres with grass or other forage growth. Forest forage growth consists of forest, brush-grown pasture, arid woodlands, etc. The total acreage of forested grazing land includes woodland pasture in farms plus estimates of forested grazing land not in farms. The data on active grazing is usually obtained from the Census of Agriculture, the NRI, and the U.S. Forest Service. |
| | Forest land not grazed | Forest-use land which is not used for grazing. |

**Table 1.** *Cont.*

| Land Type | Categories | Explanation |
|---|---|---|
| Cropland | Definition | The combined data for cropland harvested, crop failure, and cultivated summer fallow is known as cropland used for crops, or the land used as an input to crop production. |
| | Cropland harvested | Row crops and closely sown crops like Wheat, Soybean, Potato, Sugar beets, Canola, Corn, Cotton, Soybeans, etc.; Hay and silage crops like Rye, Oats, Alfalfa, Clover, Sorghum, etc.; Tree fruits, small fruits, berries, and tree nuts like Apple, Oranges, Almonds, Grapes, Blue berries, etc.; Vegetables and melons, e.g., Lettuce, Tomatoes, Potatoes, Cucumbers, Cantaloupe, watermelon; Miscellaneous other minor crops. Annual estimates of cropland harvested are based on (a) census data and, (b) the series provided by National Agricultural Statistics Service (NASS) on principal crops harvested |
| | Crop failure | As the name indicates, crop failure includes the acreage with failed crop. Extreme weather conditions, insect/diseases are the major reason for the crop failure but other reasons such as land not harvested due to lack of labor, low market prices, etc. Crop failure = cropland planted − Cropland harvested. Here, we do not include the crops which are not meant to be harvested (e.g., cover crops, green manure crops, other soil improvement crops, etc.). the acreage with such crop is included in crop idle. In recent years, crops have failed on 2–3% of the acreage planted for harvest. Annual estimates of crop failure have been calculated from the data provided by NASS. |
| | Cultivated summer fallow | The acreage with crop which is cultivated and kept for one or more seasons is considered as cultivated summer fallow. It is usually grown in the sub-humid regions of the west to stock up the moisture and control the weeds before planting the small grains. This practice is mandatory for dry cropland usually found in the west. crop production in the drier cropland areas of the West. We do not include the crop which is a soil improvement crop but not harvested and left idle. Since the late 1970s, Census of Agriculture and unpublished NASS data are the sources for Cultivated summer fellow. Before that it used to be calculated from various separate sources. |
| | Cropland pasture | Here, the cropland is used for pasture without any changes made or being harvested. If the acres are grazed before or after harvesting it is not considered as cropland pasture. Estimates were derived from the census of agriculture. |
| | Idle cropland | It includes the acres without any kind of plantation without the aim to be harvested at some point like cover crops or soil improvement crops. Various physical and economic reasons also play a part to make a cropland considered as idle. Land which comes under Federal farm program and not being used as cropland also falls under this category. Additionally, land included in USDA's Conservation Reserve Program (CRP) and Wetlands Reserve Program (WRP) is considered as idle. |

**Table 1.** *Cont.*

| Land Type | Categories | Explanation |
|---|---|---|
| Urban land | Definition | Urban area includes residential areas and concentrations of nonresidential urban areas, such as commercial, industrial, and institutional land; office areas; urban streets and roads; major airports; urban parks and recreational areas, and other land within urban-defined areas. |
| | | Census urban areas include densely populated areas with at least 50,000 people ("urbanized areas") and densely populated areas with 2500 to 50,000 people ("urban clusters") Densely populated areas-blocks with a population density of at least 1000 people per square mile, surrounding blocks with a density of at least 500 people per square mile, and "less densely settled blocks that form enclaves or indentations or are used to disconnect discontinuous areas with qualifying densities." The Census Bureau collects and prepare the data for urban area every 10 years. Here, cities that are essentially rural in character are excluded. |
| Grassland and Pasture | Definition | All open land, regardless of the ownership, used primarily for pasture and grazing falls under this category. |
| | | It includes shrub and brush land types of pasture, grazing land with sagebrush and scattered mesquite, and all tame and native grasses, legumes, and other forage used for pasture or grazing. Usually, grassland pasture and range are not always clearly distinguish able from other types of pasture and range and they almost always overlap in transitional areas |
| Special Use Area | Definition | Land which is mainly used for miscellaneous purpose which bears farmsteads, roads, lanes, etc. is considered as special use land. |
| | Rural transportation | It includes roads, railway lines in rural areas, highways, airports and all the various types of transportation methods outside of urban areas. |
| | Rural parks and wildlife areas | As the name indicates, this type of land includes wilderness areas including Federal and State parks. |
| | National defense and industrial areas | These are mainly used for military purpose which includes housing, airfields, R&D (Research and Development), etc. Such land is owned by Department of Defense and Department of Energy. |

Note: Information in table was taken from the USDA website for land type definitions and descriptions.

This website compiled information on land use and land cover estimates in the United States of America (USA). The datasets were obtained from various federal land monitoring and management sources shown in Table 2. These include the U.S. Forest Service (USFS), Bureau of Land Management (BLM), National Agricultural Statistics Service (NASS), Natural Resources Conservation Service (NRCS), U.S. Census Bureau, and U.S. Geological Survey (USGS). The information is contained in excel files that gave data figures in "millions of acres" in each dataset. This was carried out for each state in the USA, and it included information for (1) Cropland, (2) Forest land, (3) Special use areas, (4) Grassland pasture and range, and (5) Urban land.

*Procedure*

1. Land-type data were collected from excel files from the Economic Research Service (ERS.) USDA website, retrieved from https://www.ers.usda.gov/data-products/major-land-uses/major-land-uses/ (accessed on 8 December 2022). The data showed acreage data in the form of "millions of acres".
2. The data was then prepared and categorized into an excel file with five major land types:

- Cropland total,
- Forest land total,
- Grassland total,

- Special-use area total,
- Urban land total.

3     The land type categories comprised twelve states in the SEUS from 1945–2012: Alabama, Arkansas, Florida, Georgia, Kentucky, Louisiana, Mississippi, North Carolina, South Carolina, Tennessee, Virginia, and West Virginia.

4     The algebraic techniques of "Differencing" was applied to the data to get desired results for each time period (1949–2012). Data analysis by differencing involved subtracting acreage data of the former time period from the latter time period. This shows changes in acreage; the results were then plotted separately on graphs for each state [48,49].

5     The algebraic differencing formula was used to calculate the changes between the acquisition dates of the two time periods, $X_1$ and $X_2$. Therefore, the formula used was [50]:

$$X_2 - X_1 = X_d \tag{1}$$

6.     Line graphs were plotted for differencing results for each state for the analysis, these graphs include:

- Plots for the five land types showing the time period,
- Each time period is separated by five-year intervals (1949–2012),
- Comparative line graphs of all the states were carried out for the major land types, comparing the land types for differencing.

This will be used to give an estimated change over the time period.

**Table 2.** The principal data and sources for land type assessment/analysis.

| Data Types | Land Type | Data Set, Descriptions, and Link | Assessed | Last Update |
|---|---|---|---|---|
| Acreage | Forest land | Total forest-use land, 1945–2012, by State: Forest-use land grazed and forest-use land not grazed. | 8 December 2022 | 28 August 2017 |
|  | Cropland | Cropland, 1945–2012, by State: The sum of cropland used for crops, cropland idled, and cropland used for pasture. |  |  |
|  | Grassland | Grassland pasture and range, 1945–2012, by State: Grassland and other nonforested pasture and range in farms plus estimates of open or nonforested grazing land not in farms. Does not include cropland used for pasture or forest land grazed. |  |  |
|  | Special Use land | Total special uses, 1945–2012, by State: Rural transportation, rural parks and wildlife, defense and industrial, plus miscellaneous farm and other special uses. |  |  |
|  | Urban land | Urban area, 1945–2012, by State: Densely-populated areas with at least 50,000 people (urbanized areas) and densely-populated areas with 2500 to 50,000 people (urban clusters) |  |  |

## 4. Results

### 4.1. Land-Type Data Analysis

There were various land type changes to address this study's first objective. Figure 2a–l shows significant land type changes over the period (1945–2012) among the twelve states in the SEUS. While the graphs have not demonstrated significant differences between the years within each state, some states have significant outliers. When comparisons were made between states for the various land types, apparent differences indicated exponential growth in these states' landscape changes. Land type changes over each time period were

compared to previous years and showed where there were increases and decreases in acreage. This interpretation indicated one land type to the other, and acreage would have shifted among the land types. A comprehensive observation of the findings from this analysis showed that two land types (Forest land and cropland) accounted for 55% of the data set when compared to the other land types. There was an estimated average forest land change of 12% over the time period; forest land fluctuated from 1945–2012 according to the ERS dataset. Agricultural land cover for the period is estimated to be 20% according to agricultural census data 1940–2012 (https://agcensus.library.cornell.edu/ accessed on 8 December 2022), and the average population grew significantly from 1950–2012 at a rate of 2.59% annually according to U.S. Census in the region (https://www.census.gov/data/tables/time-series/dec/popchange-data-text.html accessed on 8 December 2022). This would have caused significant changes in land conversion and conservation over time. Additionally, other general observations include:

- In general, some states had significant outliers for forest land and cropland, while the other land types were stable throughout the period.
- Urban land use total showed increased towards the end of the time period. The leading driver of this change has been attributed to rapid population growth in the SEUS [11].
- Grassland and special-use areas fluctuated based on how forest land and cropland acreage fluctuated.
- Special-use area total remained stable throughout the time period for most of the states and, in some cases, responded to urban land increases.
- Cropland acreage decreased after the year 2000 for most of the states.
- The population increased for most states; Florida and Georgia stand out with greater urban sprawls.

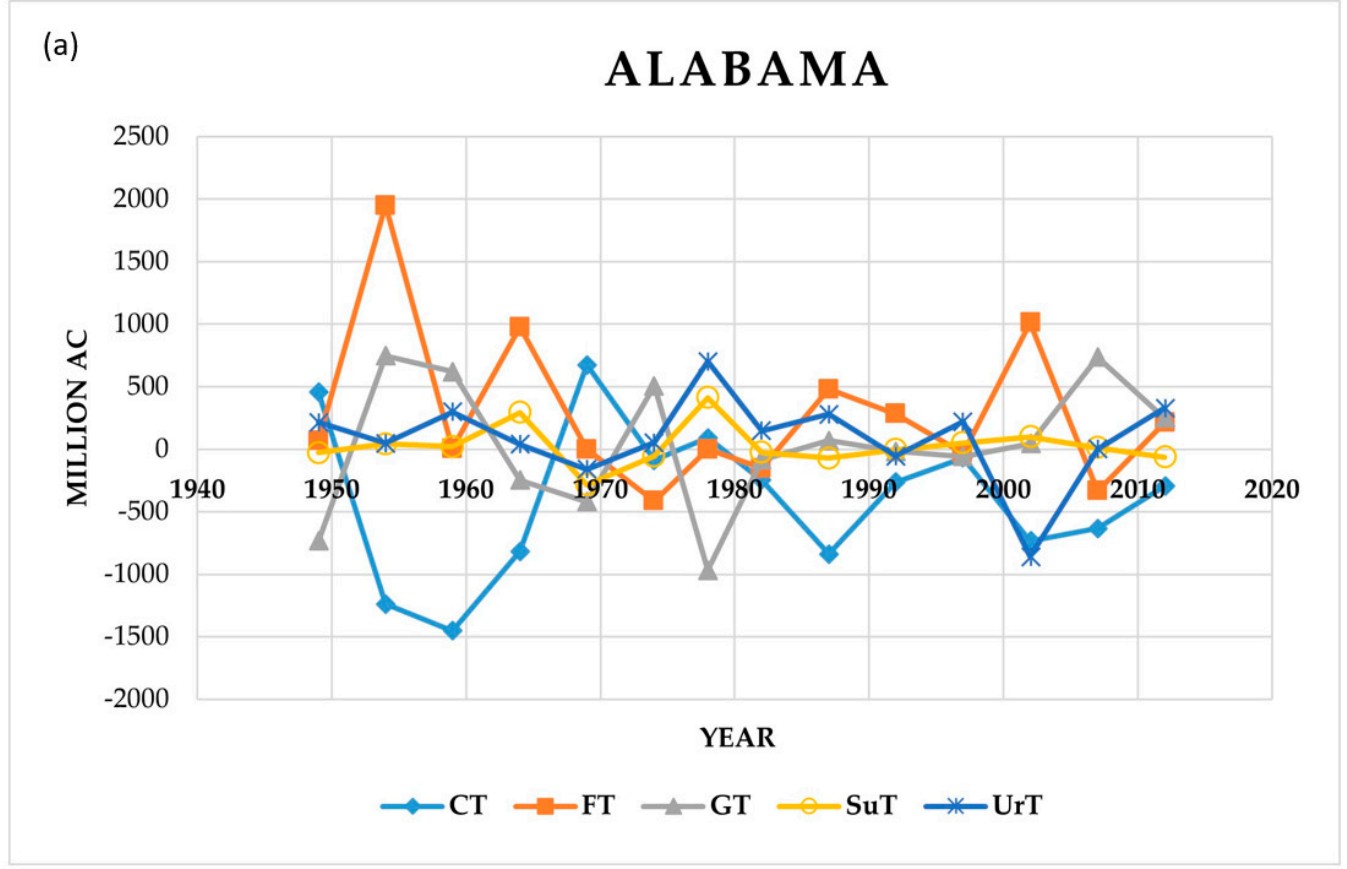

**Figure 2.** *Cont.*

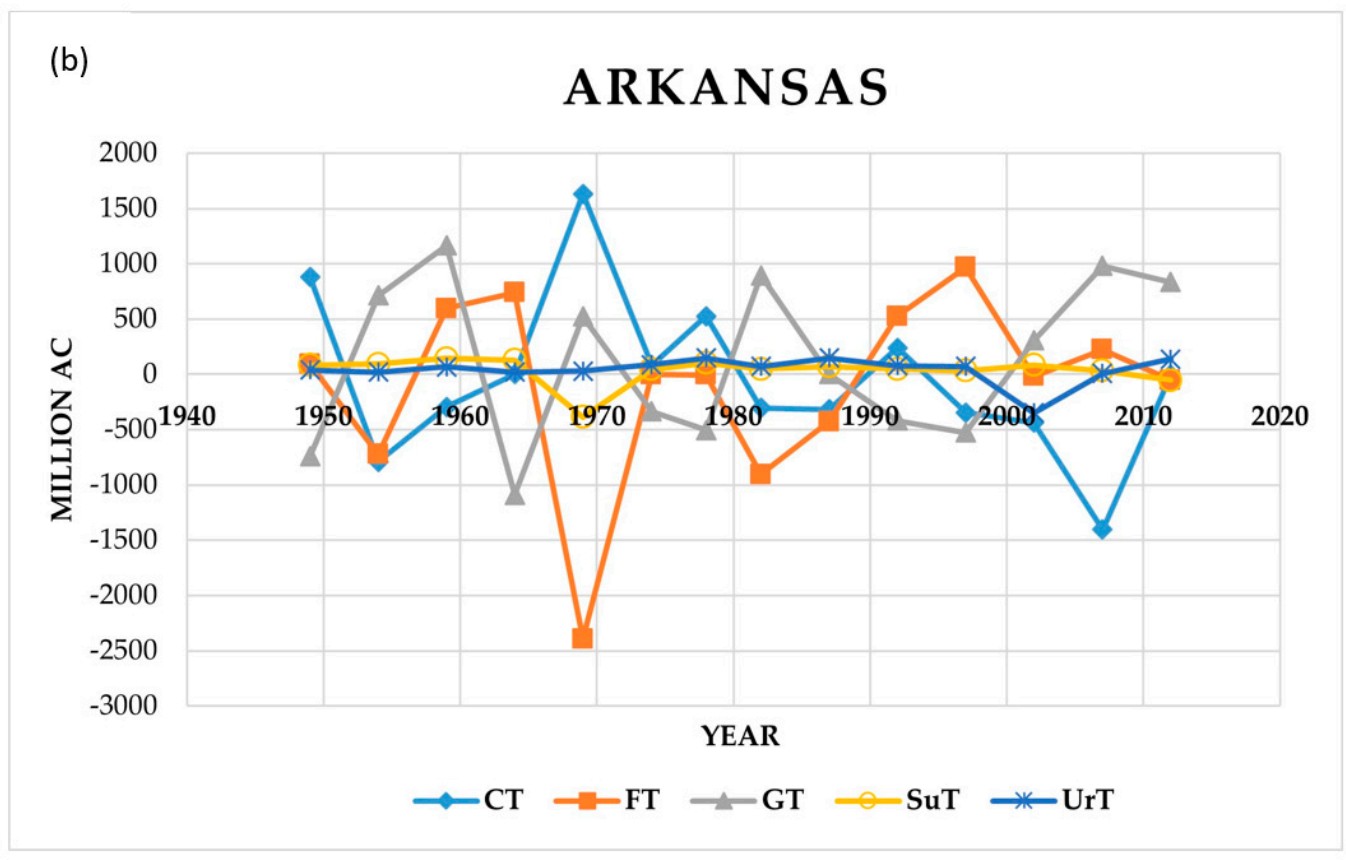

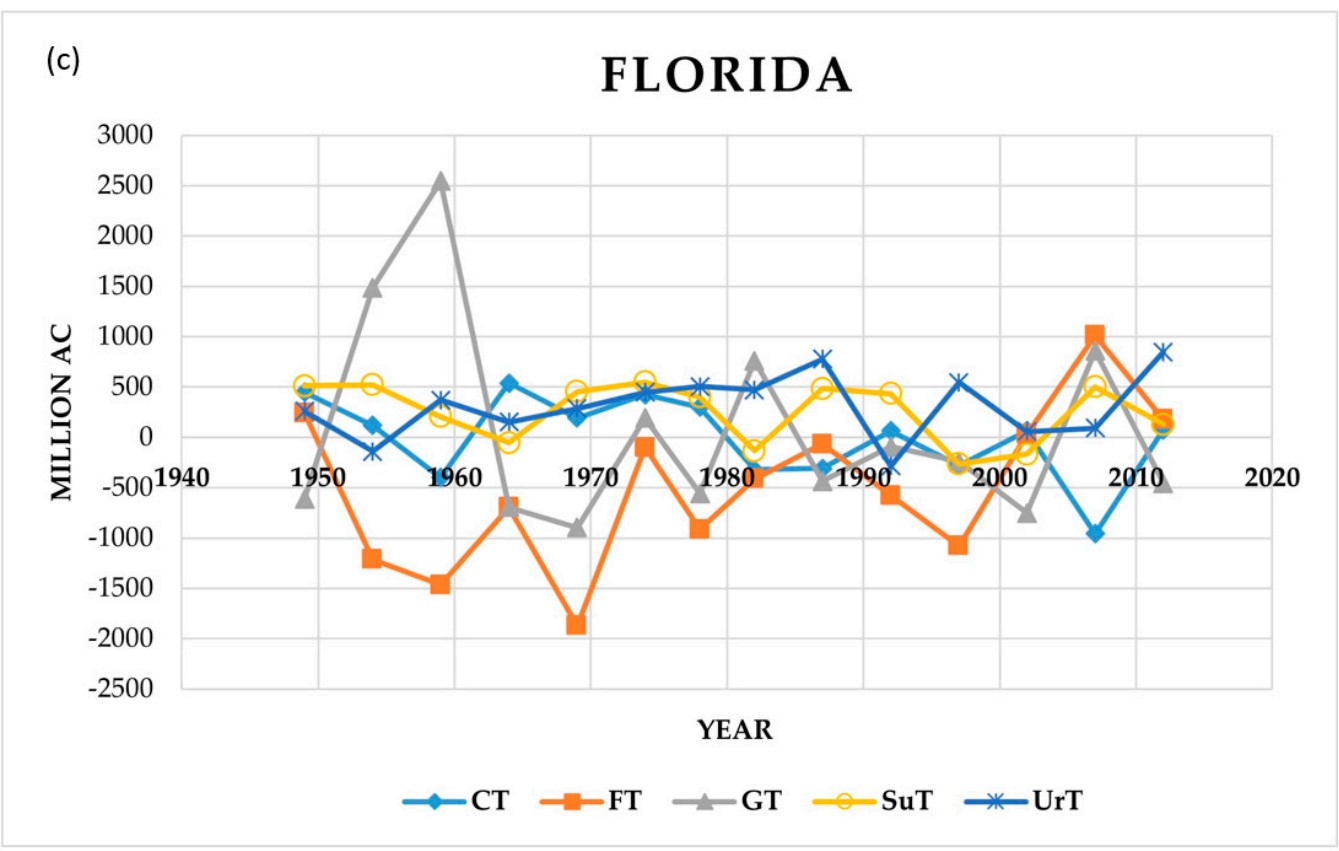

**Figure 2.** *Cont.*

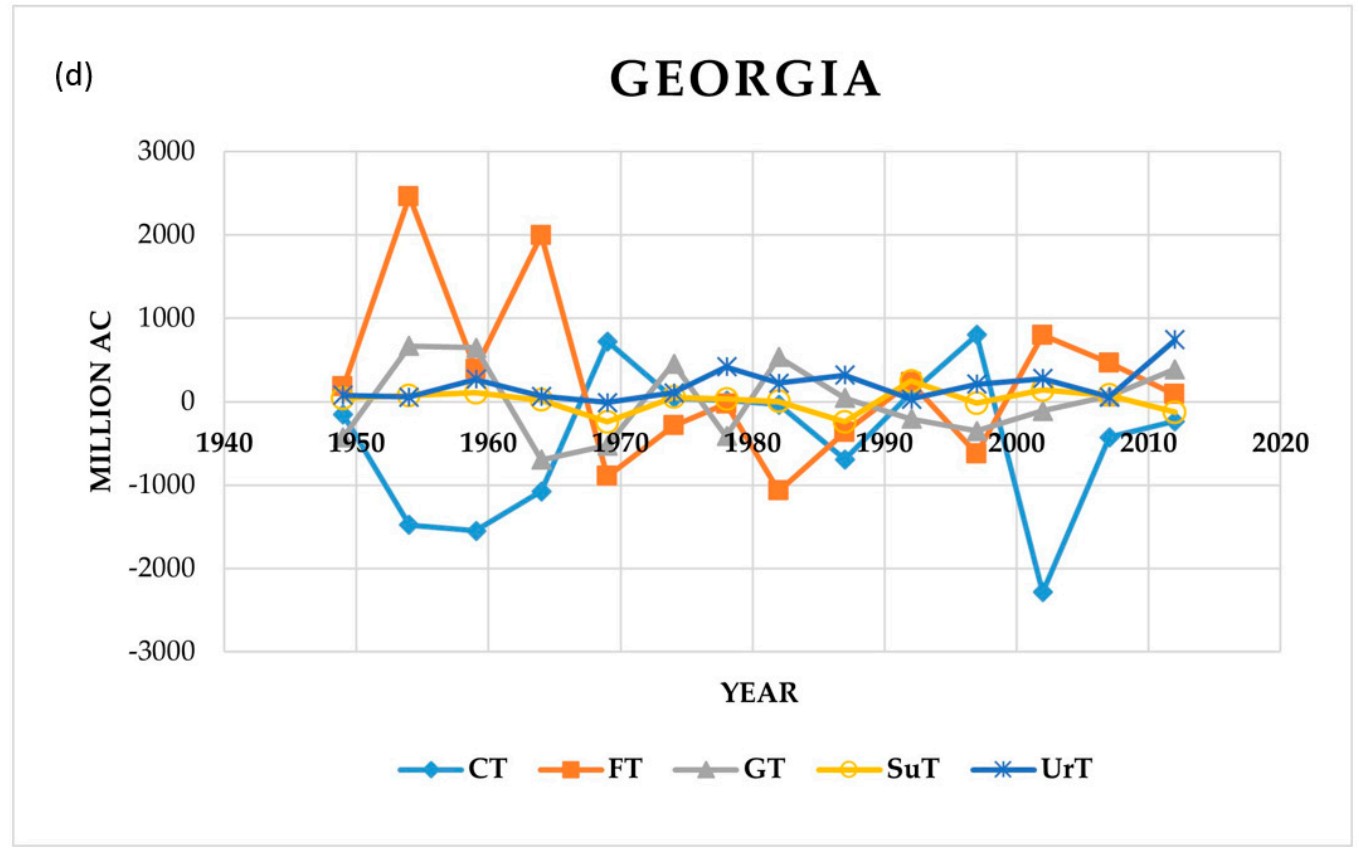

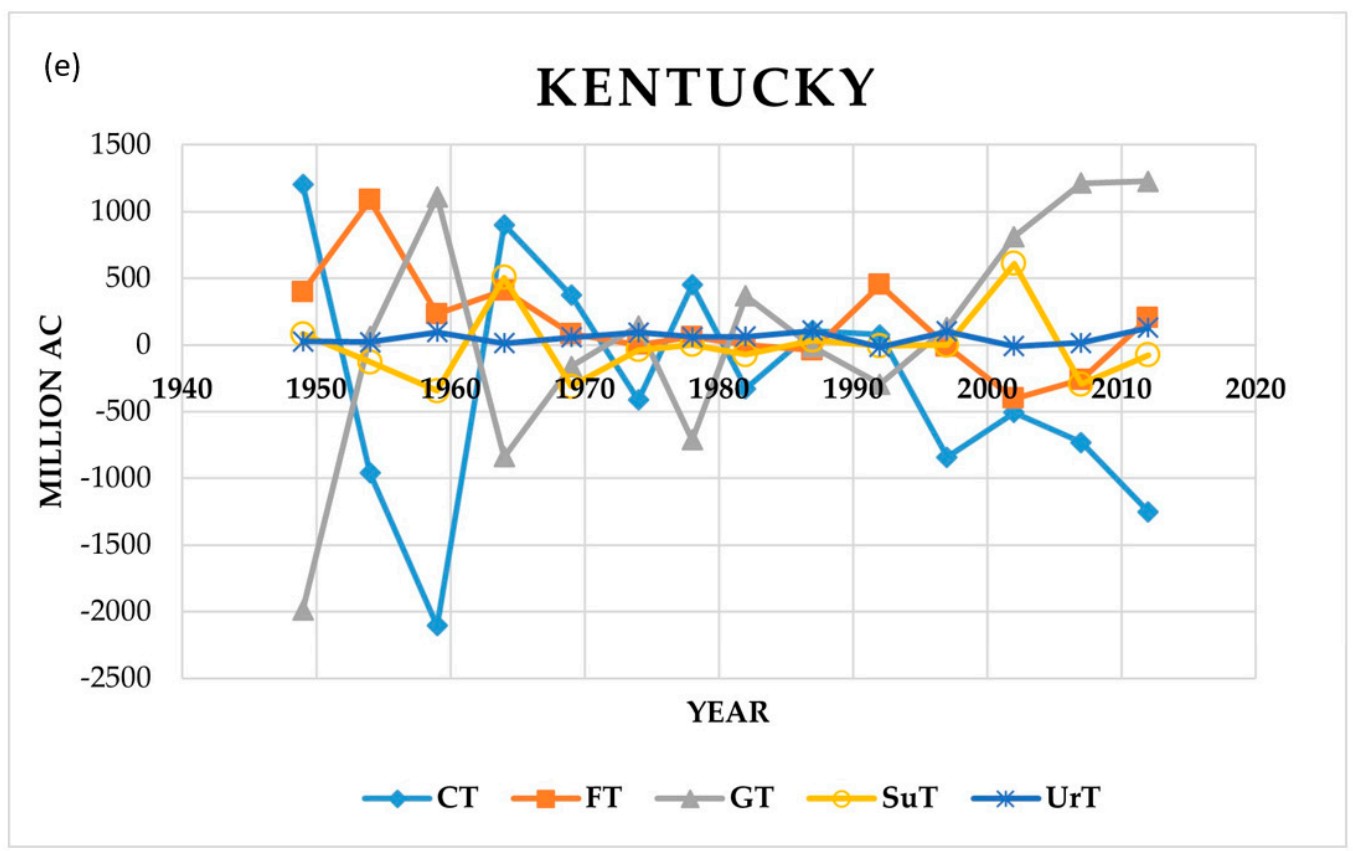

**Figure 2.** *Cont.*

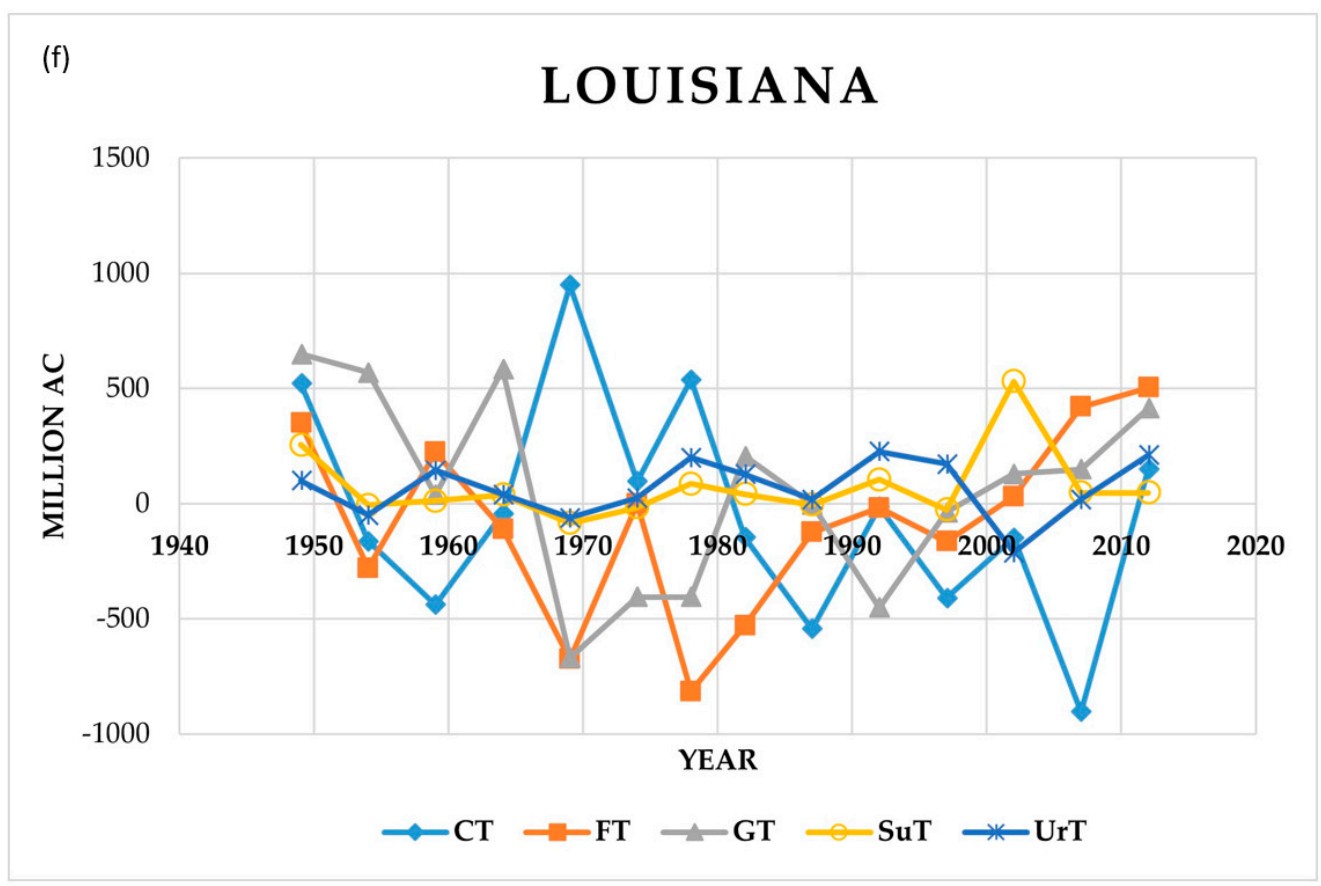

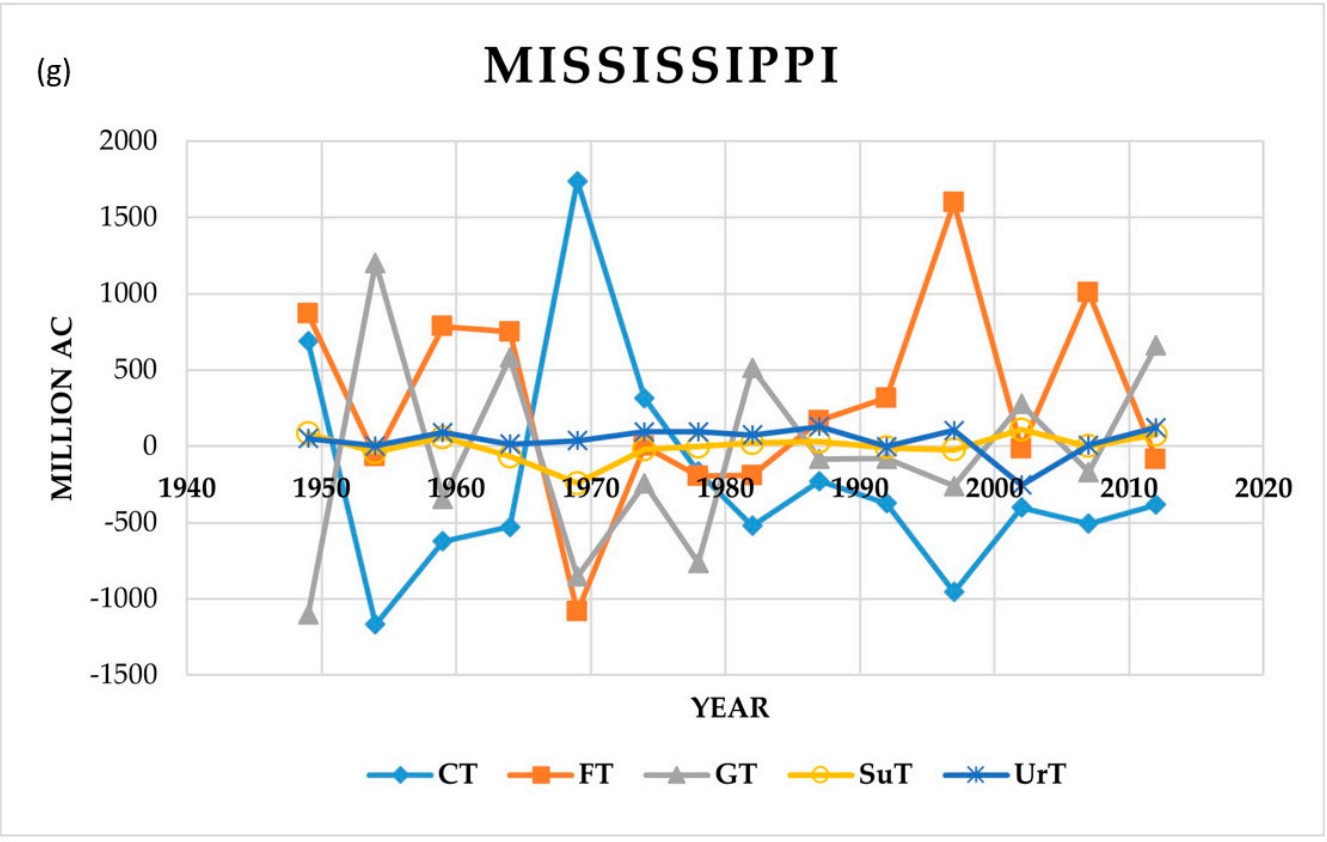

**Figure 2.** *Cont.*

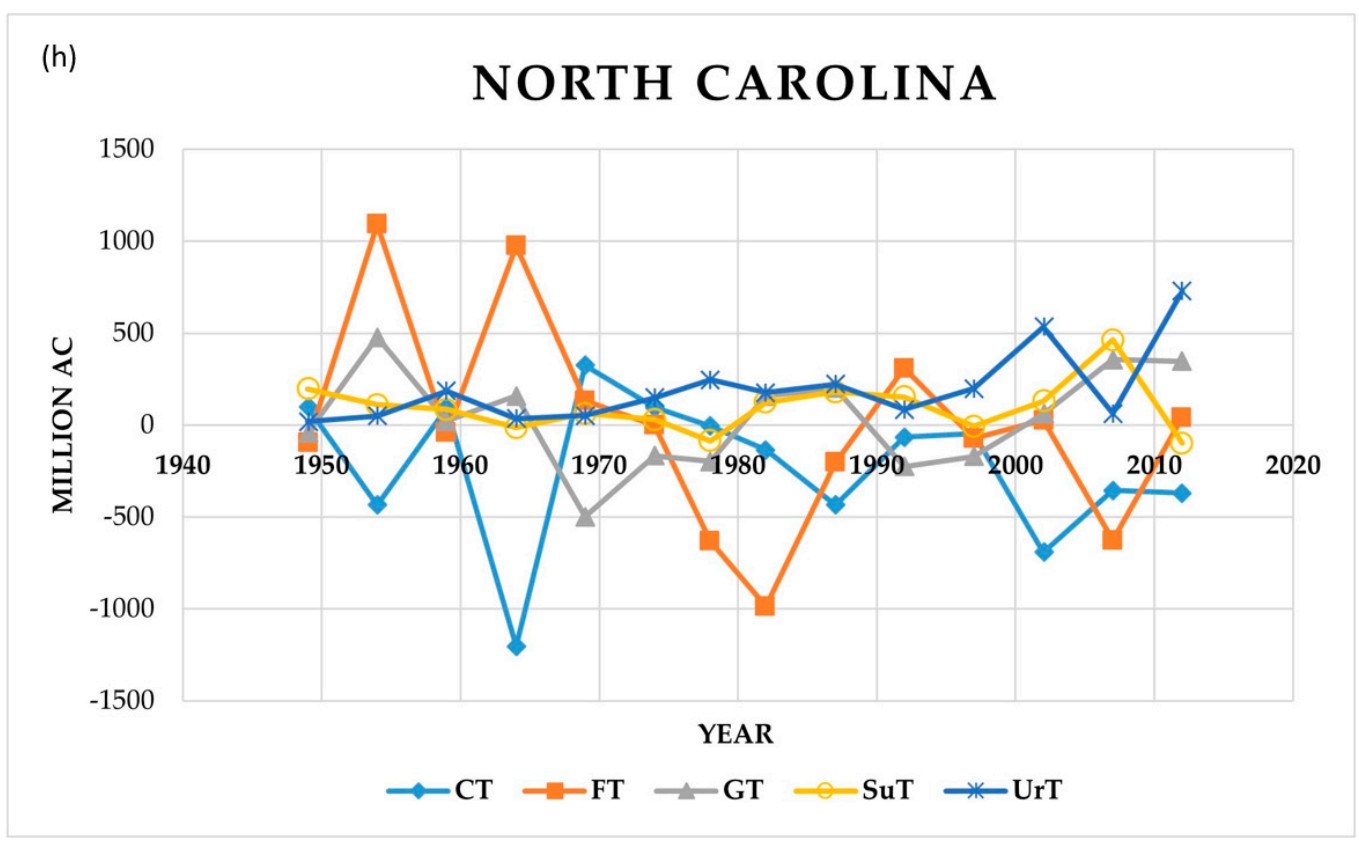

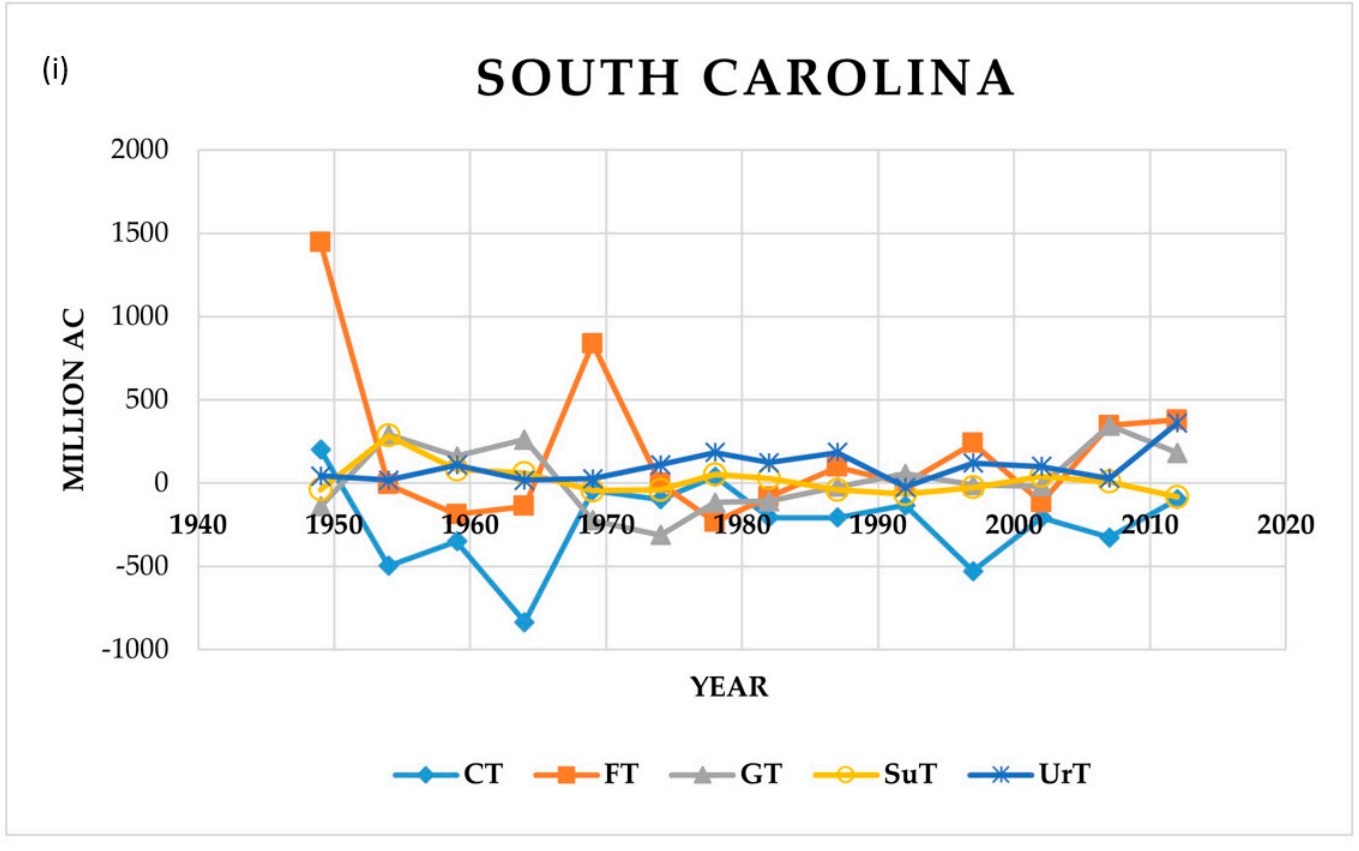

**Figure 2.** *Cont.*

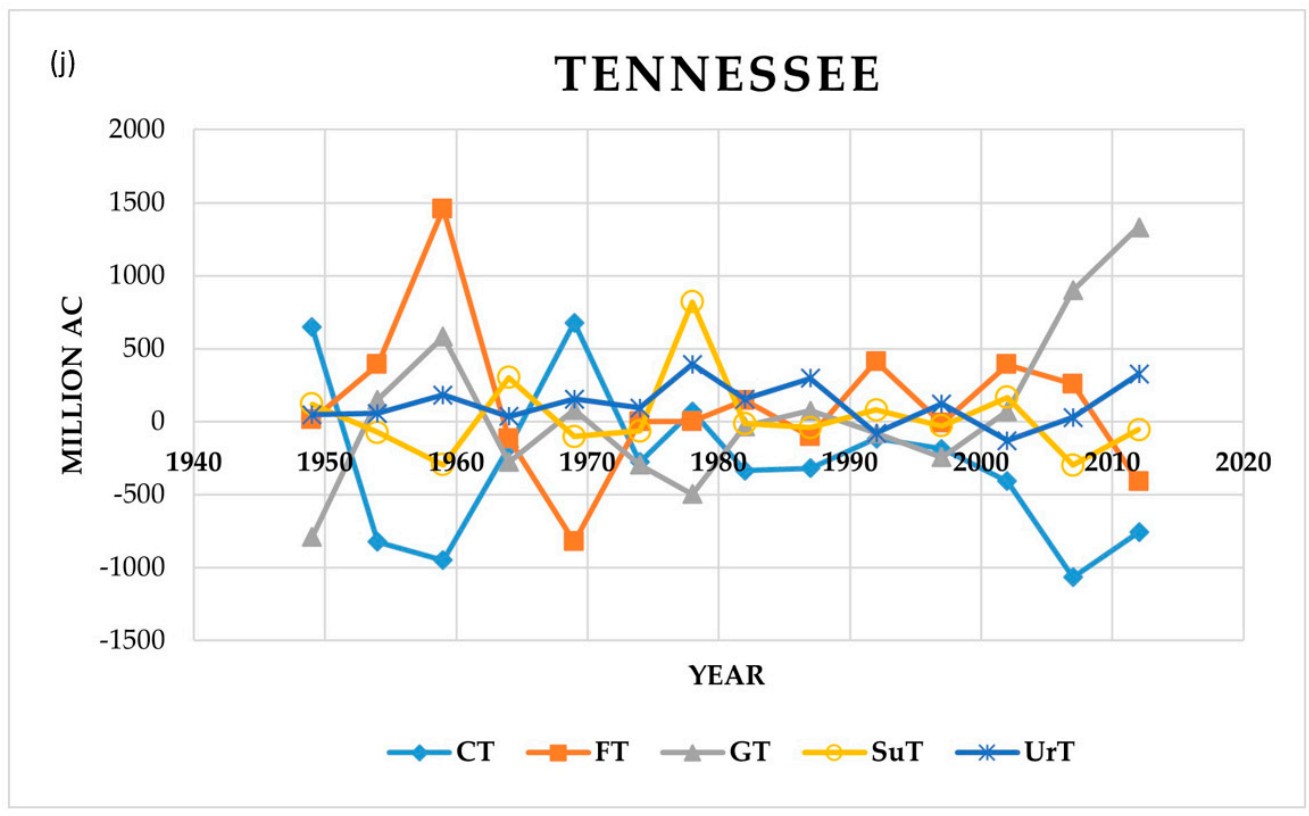

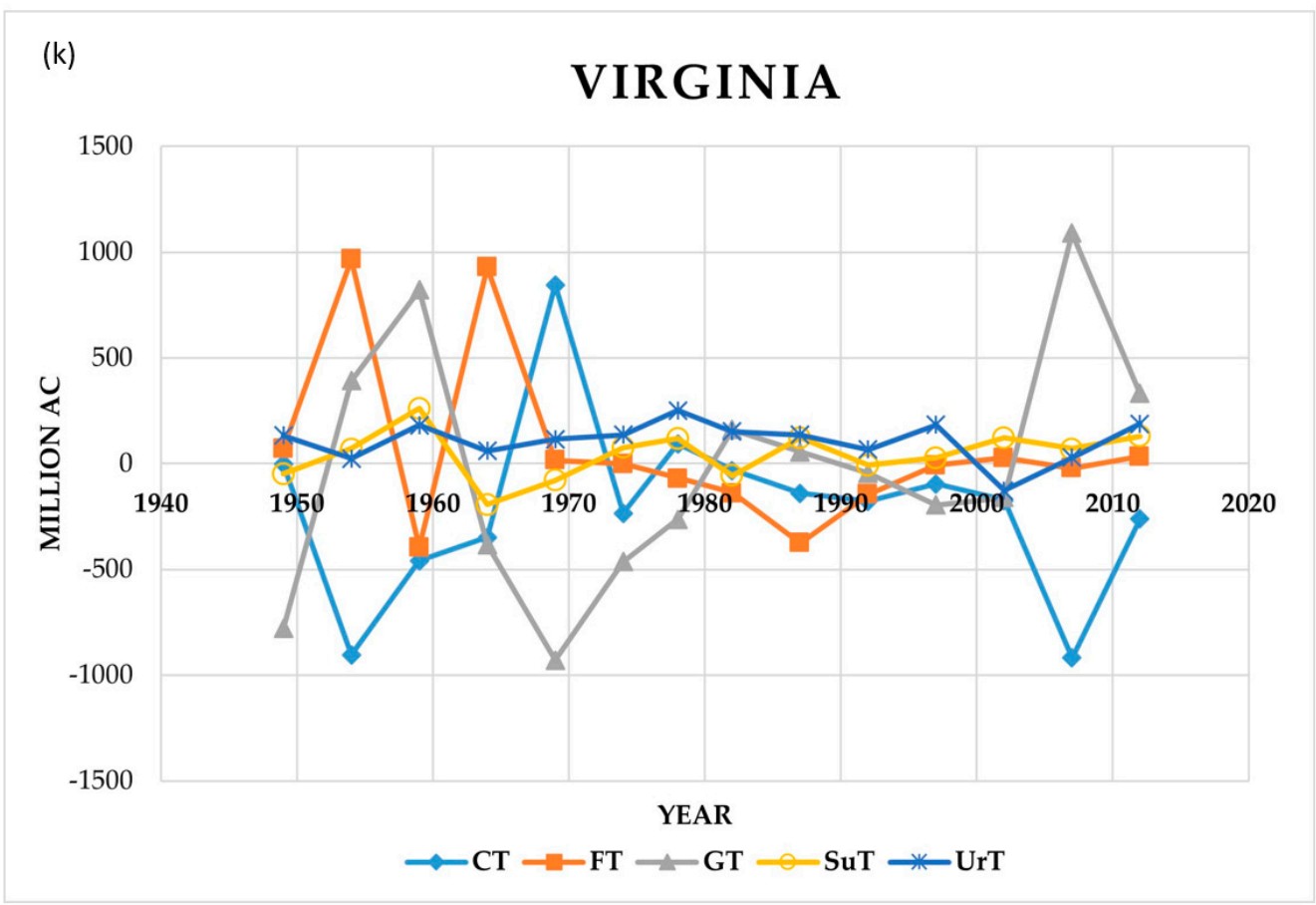

**Figure 2.** *Cont.*

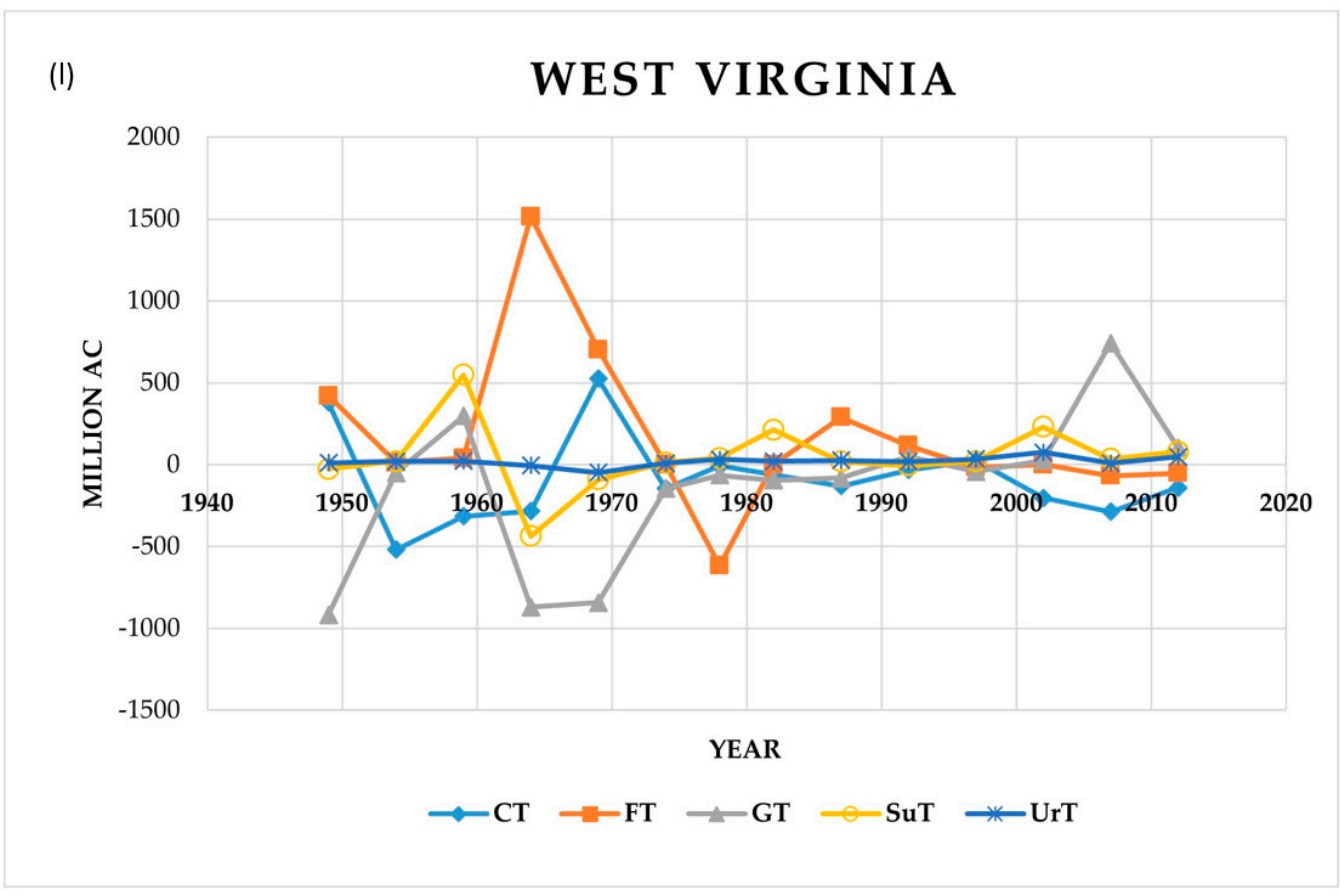

Title: Legends

**FT** - Forest land Total
**CT** - Cropland Total
**GT** - Grassland Total
**SuT** - Special-use Total
**UrT** - Urban land Total

**Figure 2.** Land type analysis of twelve states for five land types in the SEUS (**a**–**l**).

*4.2. Statewide LULCC Analysis*

The following gives some critical observations for each state in our analysis.

(a) Alabama showed Forest land increased from the 1950s to 1960s in response to decreased cropland and urban land fluctuations from the 1950s to the 1970s. All the land types were stable and within range for the remainder of the period (e.g., 1970–2012). However, there was an increase in forest land in the 2000s while urban and cropland decreased.

(b) Arkansas showed a significant cropland decrease in the 1950s, an increase in the 1970s, and then it stabilized until the 2000s. Forest land showed a steep decline in the 1970s as opposed to cropland increase; however, there was an increase from the 1980s, when cropland and urban land increased. The other land types in the analysis fluctuated over time, which could have accounted for cropland, urban land increase, and forest decrease.

(c) Florida showed a significant increase in grassland from the 1950s to 1960s, then decreases from 1960 to 1970 that showed the increase in other land types, more specifically urban land types. Forest land was also reduced in this period (1960–1970s) in response to urban land increases. Urban land significantly increased during the 1960s to 1970s, and it fluctuated until there was another increase from the 1990s to 2000s and a decline from 2000 to 2010. Overall, this state had one of the highest increases in urban land use, according to the data.

(d) Georgia showed cropland decreased from 1950–1960s. However, there was a slight increase in the 1970s, which stabilized in the 1990s, and a steep decrease in the 2000s for this land type. There was an increase in cropland from 2005, which shows an increase from 2007 to 2012. Forest land started with a steep increase in the 1950s and 1960s; however, it was stable for the rest of the time period. Urban land type increased significantly in this state after the 2000s, while the other land types fluctuated based on the response to forest land.

(e) Kentucky showed cropland acreage decreased significantly from the 1940s to the end of the1950s. However, there was an increase in the 1960s and fluctuated throughout the other time period. Grassland increased significantly from the 1940s to the end of the 1950s as a result of cropland decreasing. There was a decrease in the 1960s, which fluctuated until 2000s, and increased in 2012 in response to the decline in cropland. The other land times fluctuated over the time period but were all in response to cropland and grassland.

(f) Louisiana showed grassland, and special-use land fluctuated throughout the period, along with cropland; although there was a steep decline in the 2000s, there was a sharp increase after 2010. Forest land showed fluctuation in response to cropland and grassland changes. Urban land type showed a steep rise in 1950 and fluctuated over the period; however, there was a significant increase in the 2000s.

(g) Mississippi showed high amounts of fluctuation with all the land types. Cropland showed decreases in the 1950s, and a steep increase in the 1970s, with stable patterns of decline from 1990 to 2010. Forest land and grassland decreased in the 1970s, there was a significant increase in forest land from 1980s–1995, and a decline in 2000, and 2010. Urban and special-use land areas were stable throughout the period, all in response to forest and cropland.

(h) North Carolina showed forest land fluctuated from the 1940s to 1960s, then significantly decreased from the 1970s to 1990s, then fluctuated from 2000 to 2012. Cropland had the opposite effect on forest land, with decreases when forest land increased. Urban land was stable throughout the period, significantly increasing from 2000 to 2012, relating to forestland decreasing. Grassland and special-use land remained stable throughout the period, and most were in response to forest land and urban land use.

(i) South Carolina showed Forest land started at a high peak in the 1950s but fluctuated throughout the period. Cropland started with a low peak in the 1960s, with an increase in the 1970s, and then was stable throughout the period. The other land types (Urban, Grassland, and Special-use land) were stable throughout and fluctuated a bit.

(j) Tennessee showed cropland started low in the 1940s, but there was an increase from 1950s–2010, along with the other land types. Forest land showed a significant increase in the 1960s and a decrease in the 1970s; however, it became stable throughout the period. Urban land was stable, along with grassland and special-use land, and they were in response to cropland and forest land.

(k) Virginia showed many outliers at the start of the period. Forest land fluctuated with high peaks of increase and decreases in comparison to grassland and cropland changes, and then it was stable from 1970–2010. Most of the land types were stable after the 1970s in this state. Grassland significantly increased after the year 2000, in comparison to cropland decrease in the same time period. Special-use areas showed growth from the 1950s to the 1970s, and then there was a decline from the 1980s to the 2000s.

(l) West Virginia cropland fluctuated from the 1950s to the 1970s but was stable from the 1980s to the 2000s. Forest land showed a significant increase in the 1960s, but it was stable compared to the other land types. Urban land remained stable throughout the period with no substantial peaks. Grassland, special-use areas, and urban land types were stable throughout the years.

### 4.3. Drivers Associated with Land Use Change in the SEUS

Drivers of LULCC have been identified as those artificial pressures on natural resources that affect land change and lead to emissions [51]. These drivers that have increasingly influenced the SEUS's rate of land use change must be organized and analyzed for clarity.

Their sources must be distinctive for ease of understanding. In Figure 3, our research outlined the drivers associated with LULCC in the SEUS. The drivers were further divided into natural and artificial as the major categories. There were direct and indirect drivers for each significant category associated with LULCC in the SEUS. The direct natural drivers are natural disasters that affect the physical landscape that leads to land change. According to Emrich and Cutter [52], natural disasters cause modification to the climatic zone of the SEUS for an extended period. The indirect natural drivers are morphological, which is the natural vegetation cover and use of land, depending on the interaction between humans and the environment and the direction of land change [53]. Direct artificial drivers fall under the categories of (1). Land monitoring, which involves the various processes of land use planning, and (2). Globalization is associated with population growth.

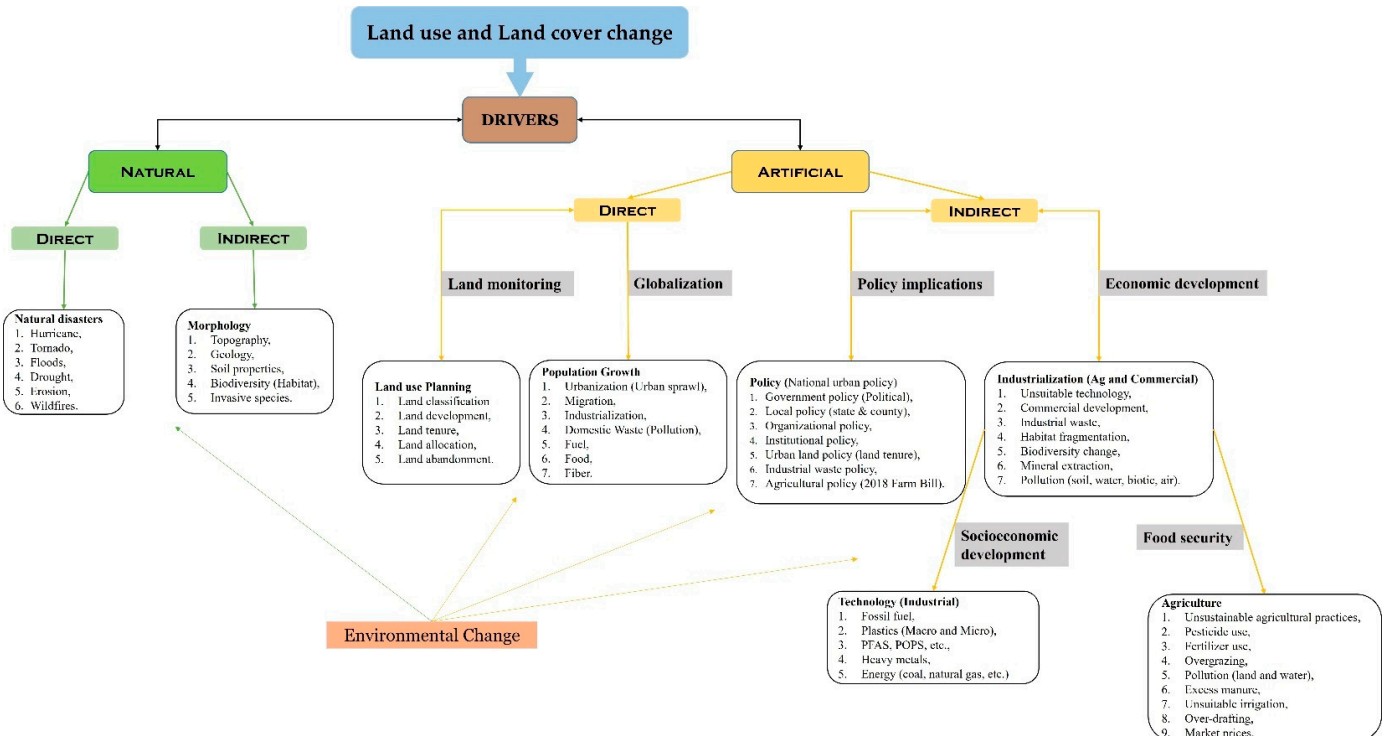

**Figure 3.** Illustration of drivers associated with LULCC in the SEUS.

The indirect artificial drivers fall under the categories of (1). Policy implications associated with the various policies surrounding land use and land cover; and (2). Economic development is further broken down into two categories: (1). Socioeconomic development, which encompasses technological advancement at the industrial level, and (2). Food security encompasses agriculture and the processes that provide food for a growing population. Land use trends and changes are elevated based on the drivers identified and the magnitude and distribution of these changes [1]. These drivers affect land use change, and their effects eventually lead to environmental change. According to Turner et al. [54], changes in land use systems are related to environmental change in an area, affecting habitat loss and altering other forms of the ecosystem. Figure 3 gives a schematic illustration of land use change drivers and the effects of long-term environmental repercussions [55].

The impact of human activities and alteration to the earth's surface (land change) has altered the environmental makeup of the SEUS; however, if appropriately managed, land change can significantly affect ecological growth and sustainable environmental growth [56]. Categorizing the effects of land use change based on the uses of specific land types can help researchers in various ways, not only for decision-making and policy changes on a federal level but for researchers to understand the interrelated dynamics, patterns, and processes associated with land use change [57]. Our findings were used to develop a schematic

illustration of the various land types, the effects of land use change, and how they lead to other processes that affect the environment (Figure 4).

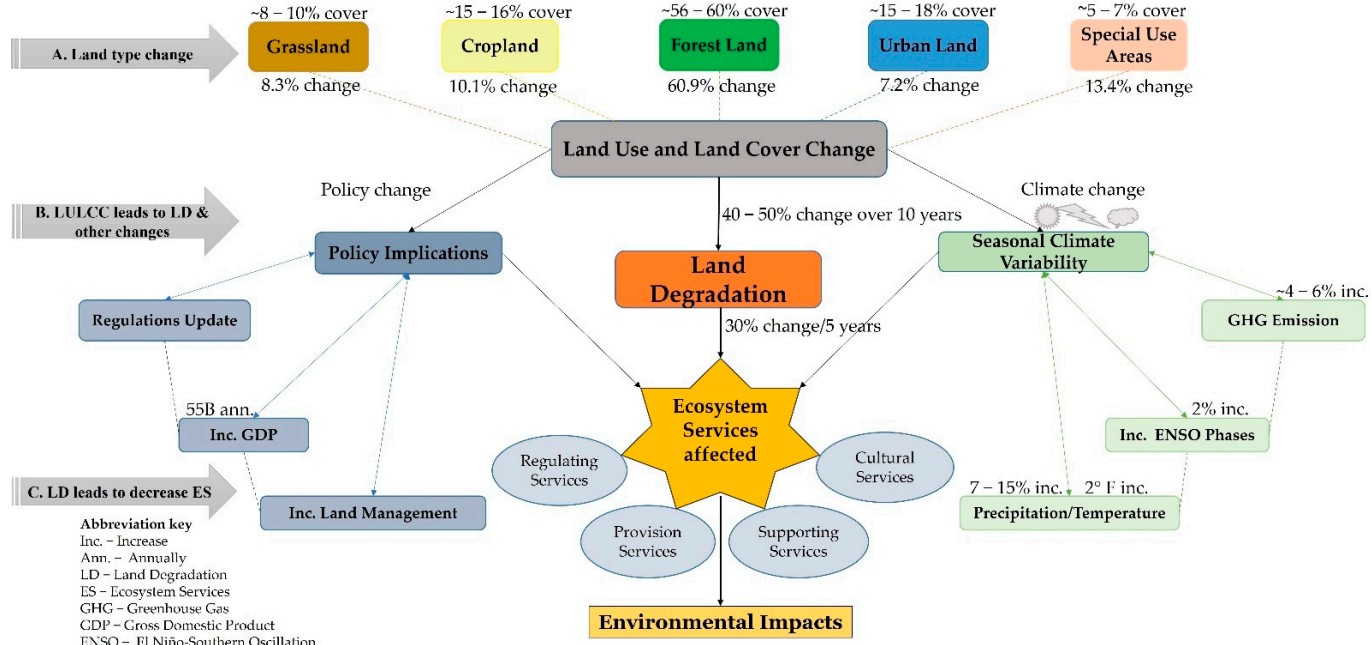

**Figure 4.** A. Schematic illustration of LULCC in the SEUS from 1945 to 2012 [13,58]; B. Land degradation changes from LULCC over ten years [13,15,59–65]; C. Land degradation changes that lead to percentage depletion of ecosystem services [66,67].

The diagram shows the coverage of the major land types and the percentage change over time. The reported changes can eventually lead to land degradation, affecting ecosystem services (Figure 4). Ellenburg et al. [58] also reported similar changes in the SEUS, categorizing the four areas of ecosystem services not functioning. The figure also shows the four categories of ecosystem services being mainly affected by LULCC at an alarming rate. These areas include provisioning, regulating, supporting, and cultural services. Nepton et al. [11] also showed the changes within the SEUS related to land changes, leading to ecosystem services being affected. Ecosystem changes have been observed through land classification over time. Additionally, the various drivers give a clear understanding of the impacts and the assessment of land degradation. Changes in seasonal climate variability have been linked to LULCC in the region [61,68]. Climate variability due to anthropogenic activities provides decision-makers with information to update current policies for land-use and monitoring in the SEUS [6,63]. This illustration shows the linkage between the various players of LULCC; however, the effects are still not accurately measured. These are percentage changes over an annual period, as such, these changes are readily occurring because of continued land use. The result has also correlated with land-use findings of other researchers in the region.

## 5. Discussion

### 5.1. Land Use Changes in Individual States

The analysis also showed some states with significant increases and decreases in cropland and forest land while others were stable regarding grassland/pasture, urban and special-use areas. This was confirmed by Asseng [15], which showed 60% for forest land, 15.7% for cropland, and 8.3% for grassland pasture and ranges of the region's land mass. The major land changes in the SEUS reported by researchers have been consistent with higher crop production. The impacts on the region have been significant concerning climate change parameters, for example, rainfall variability, drought, temperature rises, higher

emissions of $CO_2$ in the atmosphere, plant sensitivity to climate changes, soil degradation, and impacts on hydrology and water quality [9,15,69]. While it will be abrupt to generalize the effects of land use and land cover changes on the SEUS, the impacts are general for each state within the climate change parameters. Some states within our investigations have stood out based on the land use conversions from one land type to the other; for example, within the sixty-seven (67) year time span, Georgia has one of the highest peaks of urban land total growth; this was also reported by Weston et al. [70], that showed higher population densities between 1970 and 2000. Georgia and Florida have the highest rates of urban land development over time, with a 25% increase for Florida from cropland conversion [3].

The analysis showed some specific states reported significant outliers among the five land types in the study period. Forest was the primary land type observed in the region, and its conversion as a result of driving forces was reported by Nepton et al. [11] for the region. Approximately 10.05 million hectares of forest cover were harvested or cleared between 1973 and 2000 [4]. Some states stood out with significant land cover conversion; for example, Alabama showed a decrease in forest land from 1949 to 1969. However, the forest loss throughout the states gives rise to the other land types increasing, and this was further confirmed by Li et al. [36]. Who reported forest loss resulted from logging, which was not at a dangerous level [36]. In the 1970s, Arkansas increased cropland, with a steep decrease in forest land during the same period; this growth has continued over the decade, according to Yan and Edwards [71]. The population in Arkansas has grown over the years, resulting in a greater demand for food. The state of Kentucky is known as the bluegrass for its vast grassland/pasture and forest land [72]. Our assessment showed grassland as the major land type, which increased in the 1960s while cropland decreased. There was also an increase from the 1990s to 2012. While the impact of LULCC on specific states will vary, most states showed significant changes from forest land and grassland to cropland, special-use areas, and urban land use [45]. The effects of such conversion have reported environmental impacts on soil degradation and its ability to sequester carbon in the SEUS [73], watersheds, stream flow, hydrology and water quality [9,18,74,75], changing atmospheric and soil temperature [30,76], and climate change [77].

*5.2. Quantitative Land Use Changes in the SEUS*

The SEUS has been undergoing changing landscapes since the 1930s [78]. LULCC analysis for the SEUS is vital for understanding past and future land use and their impact on the region [46]. The results show that cropland and forest land are the most significant land use totals within the SEUS, reported by the ERC and other researchers [39,79,80]. The major land type cropland has been equated with large landscapes for intensive agriculture for food production in all of these states reported by Thornton et al. [81]. Agricultural production has contributed considerable economic incentives for the region; there has been a greater conversion of land for agriculture in the SEUS than in other areas [65,77]. According to Broussard and Turner [82], farms with more than 405 hectares (1000 acres) increased by 65%, and the average farm size doubled in the region. While its effects have affected the entire SEUS by changing weather patterns [68], extreme greenhouse gas emissions [83], habitat threats and loss [46,84], hydrological changes [69], ground and surface water pollution, droughts/carbon fluxes [85] and climate change [86] among other effects. The environmental impacts of agriculture on ecosystem services have seen landscapes converted back to open grassland and forest areas [39]. The decline in natural wetlands, forests, and grassland pastures has resulted in the SEUS increasing cropland and urban land totals [17]. The reshaping of the SEUS by economic forces has been a challenge for environmental agencies. While the SEUS seems to have large amounts of forest land, the effects of urban and cropland totals overwhelm forested areas. The anthropogenic impact on the landscape is observed at the habitat and hydrological levels [84].

The effect of deforestation, wetland degradation, and the interaction of climate change and agriculture have affected landscape changes drastically [87]. The challenges associated

with the rising population generate environmental impacts that affect forest resources, hydrological stream systems, and natural wetlands throughout the SEUS [9,88]. Some researchers reported the changes in the SEUS as an impact on the ecoregion and highlighted the various changes occurring due to human activity [27,34]. The environmental changes have been recorded; our results have shown how the landscape has shifted between the major land types. The region's economic productivity is a direct contribution from agriculture (cropland) and urban expansion, resulting in a decline in forest and grassland areas [89]. While the economic benefits can be measured, the environmental impacts from changing landscapes can be detrimental to ecosystem services [90]. These environmental effects have led agricultural enterprises to shift to more sustainable food production in the SEUS [91,92]. Some examples include the use of Best Management Practices (BMPs) [93], forecasting for sustainable farm management [30], and the use of forest plantations [4]. Our observation of the data indicates that more land type totals have shifted to more urban land types closer to this century. However, cropland seems to have the most effects on nonpoint source pollution in the region [94,95]). Other preservation forms and switching from cropland back to forest land have improved the region's water quality [81].

### 5.3. Drivers Associated with Land Use Change in the SEUS

Landscape changes and the major drivers involved must be understood to understand human pressure effects [96]. Over time, land use and land cover change have been associated with driving forces within the SEUS. According to Nepton et al. [11], the SEUS Land use and land cover change in the SEUS are affected mainly by natural and artificial drivers. Categorizing these drivers has been done in the past for the entire USA regarding LULCC. However, some authors have only identified drivers more specific to the SEUS because they are based on the region investigated [44]. The use of remote sensing and geospatial technology has equipped the agencies to use the Landscape Change Monitoring System (LCMS) to detect multiple LULCC changes within the SEUS [97]. The drivers identified (Figure 3) associated with LULCC are essential to understanding the changes in resources to the entire SEUS. The importance is always placed on the sustainability and maintenance of water and land resources, specifically wetlands and streams, rivers, and other natural water bodies. The direct natural drivers (e.g., Hurricanes, floods, and fires) have a greater effect on landscape changes than indirect natural drivers (e.g., Topography, habitat, and invasive species). The outcome can lead to severe impacts on the land cover in the region, effects on the region's climate, and ultimately land use in the future [7,98]. However, the artificial drivers (e.g., Population growth, agriculture, and industrialization) are considered anthropogenic. Their effects are at a greater rate and are rapidly carried out in the SEUS. This was also reported by Nepton et al. [11] relating to the drivers that affect LULCC. The direct artificial drivers (e.g., Land use planning and population growth) seem to have one of the highest impacts in the region. Our results on population growth show that the population continues to grow at 2.69 %/year, with some states (Florida) having a higher growth margin than others [45]. This increases the need for land use planning, monitoring, and management in the region, which sees more landscape being converted from one land type to another. The indirect artificial drivers (e.g., Policy and economic development) are also important. The decisions around these drivers give rise to land use resolutions, and the impacts go beyond landscape changes [99]. These decisions affect the outcome of land use for other artificial drivers (e.g., technology and agriculture) that have shown significant impact on the environment. Agriculture is identified closely to urbanization because its effects affect the ecosystem [67], hydrology, and habitat [41]. These drivers allude that resources are not only essential for environmental processes but economic (agricultural) and municipal (urban) uses [89,100]. There are many economic incentives earned within the SEUS from the various land type uses, for example, forest land, although over the decades, forest clearing for other land cover types have been extensive [101]. There is always a need to understand land cover dynamics after forest clearing. As such, land cover classification is essential and provides information for prioritizing and conservation programs [38]. While

these drivers are classified into direct and indirect natural and artificial drivers, the most potent drivers associated with LULCC are the indirect drivers because of impacts from anthropogenic activities. Many other researchers have reported similar findings relating to the driving forces of LULCC [11,17,102]. Our results help illustrate land cover changes and their overall environmental effects (Figure 4). That continuously shows the various parts of the system and how they are all connected to give a baseline of negative environmental impacts [39]. While the illustration demonstrates a general overview of the environmental players involved in ecosystem services being affected, additional assessment of LULCC can determine each aspect of the entire illustration.

*5.4. Assumptions, Limitations, Next Steps*

This research used quantitative data and values from the ERS 2012 report that is assumed to be investigated by various federal agencies. Additionally, only the totals of the five major land types were used in this research out of eleven that give specific information for each area. The dataset used for this research was secondary, temporal data. The information on the dataset shared an accuracy of 80%. The methodology used was "differencing". Other methods can be used to assess the data; however, we focused on a simple technique for this data type. The drivers identified are linked to land use change in the south; however, to give the specific quantitative effect of each driver was not quantified. Therefore, the next step for this research involves understanding land use change at the degraded level and its impact on specific aspects of the ecosystem. Landscape changes are inevitable; therefore, understanding land degradation processes will be a vital asset to the topic of land use change. Spatial data analysis will be the next step in land use change and data analysis. The results from the effects can be quantified, providing researchers, land users, and decision-makers with information for informed decisions related to land use.

## 6. Conclusions

This research considers the use of quantitative data for assessment to determine land use change among five land types in the SEUS. The first objectives of this study were to show changes among the five major land types for twelve states in the SEUS using acreage data. Using a simple method of analysis (Differencing), the results showed forest land and cropland having the majority of the land cover during the time period (55%). Forest land accounted for 12% change and agricultural land for 20%. These changes were in response to the growing population in the region, which has increased by 2.59% on average annually. The land type changes were shown to increase and decrease for each time frame in the study period, and these changes were related to various land use change drivers. Objective 2 showed the drivers associated with LULCC that influence environmental change in the region. The drivers were divided into natural and artificial drivers, with eight categories showing how they affect landscape changes in the region. Finally, an illustration diagram shows the various quantitative changes within the period and how the interactions between the key players of LULCC lead to environmental change. The results concluded that land use change would continue to be a factor because of urban land development, and agriculture is the major driver. These changes will continue to cause the exploitation of land resources, triggering unsustainable land management and land policy decisions more focused on economic gains. The results from this study can be used for better decision-making related to future land use in the SEUS.

**Author Contributions:** R.N. and A.A.: Conceptualization, Visualization, Methodology. A.A.: Resources, Supervision, Project administration, Funding acquisition. R.N.: Data curation, Writing—original draft preparation. R.N. and A.A.: Writing—review and editing. All authors have read and agreed to the published version of the manuscript.

**Funding:** This research is funded by the National Institute of Food and Agriculture of the United States Department of Agriculture (USDA-NIFA) to Florida A&M University through Non-Assistance Cooperative Agreement grant no. 58-6066-1-044. Additionally, support from USDA-NIFA capacity

building grants 2017–38821–26405 and 2022-38821-37522, USDA-NIFA Evans-Allen Project, Grant 11979180/2016–01711, USDA-NIFA grant no. 2018–68002–27920, and National Science Foundation Grant no. 1735235 awarded as part of the National Science Foundation Research Traineeship and Grant no. 2123440.

**Data Availability Statement:** https://www.ers.usda.gov/about-ers/partnerships/strengthening-statistics-through-the-icars/land-use-and-land-cover-estimates-for-the-united-states/ (accessed on 8 December 2022); https://www.ers.usda.gov/data-products/major-land-uses/major-land-uses/ (accessed 8 December 2022); https://www.ers.usda.gov/data-products/major-land-uses/maps-and-state-rankings-of-major-land-uses/ (accessed on 8 December 2022).

**Acknowledgments:** The authors would like to acknowledge members of the research lab: Almando Morain, Herbert Franklin, Rahmah Alhashim, and Ernesta Hunter for editing support. The support from Katie White and Jahlita Janeau in obtaining literature and preliminary editing is acknowledged.

**Conflicts of Interest:** The authors declare that they have no known competing financial interests or personal relationships that could have appeared to influence the work reported in this paper.

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
