# Peer review of "Land Use Changes in the Southeastern United States: Quantitative Changes, Drivers, and Expected Environmental Impacts"

_land, doi:10.3390/land11122246_

Round 1
Reviewer 1 Report
In the study, land use change in a region was evaluated from different perspectives. The data set used in the study is sufficient. Although it is a good study in general, there are a few shortcomings in the presentation of the study.
In the first study, why did you divide land types into 5 major categories? Did the work zone require flour?
In the second study, the algelaraia-based technique was used for the analysis of major land types. The reason why this method is preferred, a brief information about other methods in the literature should be added.
3. The methodology part should be explained in a little more detail. Is there a checkpoint or not?
4. The general layout of the article should definitely be rearranged. The Literature Review section should be included in the introduction.
5.Figure 1 v2 2 should also be editing. Is it possible to show these changes with different graphs? As it stands, it is very confusing and unreadable. Also, abbreviations such as FT, CT are given under figure 2, but I think they should be added as legends anyway. The shapes definitely need to be improved.
6. You need to generalize the conclusion a little more. The overall results of the study should be interpreted briefly.
Author Response
In the study, land use change in a region was evaluated from different perspectives. The data set used in the study is sufficient. Although it is a good study in general, there are a few shortcomings in the presentation of the study.
In the first study, why did you divide land types into 5 major categories? Did the work zone require flour?
The separation into five categories was a better representation of the data, since these categories give a holistic view of the major land use changes in the USA, also the datasets presented by the Economic Research Service fall within each of the five categories. Our thoughts centered around making the data specific to the land types we wanted to focus on. https://www.ers.usda.gov/data-products/major-land-uses/maps-and-state-rankings-of-major-land-uses
In the second study, the algelaraia-based technique was used for the analysis of major land types. The reason why this method is preferred, a brief information about other methods in the literature should be added.
This study uses land use analysis using differencing and the systematic literature review method for acquiring articles for the driver. There were two methods of analysis for the research, we have modified the method section to clarify this in the manuscript.
Noted. Thanks for your comment
Some methods of analysis involve change detection techniques (E.g., Ratioing, princi-pal component analysis, and change vector analysis) of various datasets, maps, and satellite images to determine land use change [40]. Our goal is to use a simple algebraic technique of analysisanalysis technique known as "differencing" to show changes in land type over the period. This simple method works well with acreage data and can show the differences between the time frames.
- The methodology part should be explained in a little more detail. Is there a checkpoint or not?
The methodology was modified, the checkpoint is to show the land type changes and estimate the forest land cover, agriculture, and population growth since they are the main drivers associated with LULCC in the region.
- The general layout of the article should definitely be rearranged. The Literature Review section should be included in the introduction.
Noted. Thanks for your comment.
A brief overview was mentioned in the introduction and literature review was modified, the objectives were change and the manuscript is more focused and specific.
- Figure 1 v2 2 should also be editing. Is it possible to show these changes with different graphs? As it stands, it is very confusing and unreadable. Also, abbreviations such as FT, CT are given under figure 2, but I think they should be added as legends anyway. The shapes definitely need to be improved.
Noted, thanks for your comment.
Figure 2 (a-l) were modified to give more visibility to the legends and also how they were use. The reason for using line graphs was to show change over time, also to show increases and decreases in the land type change throughout the time period. The idea is to get the readers to understand how the five land types in each state changed (increase and decrease) from 1945-2012, therefore line graphs have proven to be the best option for the dataset. We created a separate key legend because we didn't want the graphs to look clustered.
- You need to generalize the conclusion a little more. The overall results of the study should be interpreted briefly.
Noted, thanks for your comment
This research considers the use of quantitative data for assessment to determine land use change among five land types in the SEUS. The first objectives of this study were to show changes among the five major land types for twelve states in the SEUS using acreage data. Using a simple method of analysis (Differencing), the results showed forest land and cropland having the majority of the land cover during the time period (55%). Forest land accounted for 12% change and agricultural land for 20%. These changes were in response to the growing population in the region, which increases on average 2.59% annually. The land type changes were shown to increase and decrease for each time frame in the study period, and these changes were related to various land use change drivers. Objective 2 showed the drivers associated with LULCC that influence environmental change in the region. The drivers were divided into natural and artificial drivers, with eight categories showing how they affect landscape changes in the region. Finally, an illustration diagram shows the various quantitative changes within the time period and how the interactions between the key players of LULCC lead to environmental change. The results concluded that land use change would continue to be a factor because of urban land development, and agriculture is the major driver. These changes will continue to cause the exploitation of land resources, triggering unsustainable land management and land policy decisions more focused on economic gains. The results from this study can be used for better decision-making related to future land use, conservation, and restoration of land covers in protected areas of the SEUS.

Reviewer 2 Report
This is a good topic and of interest to readers of Land. But the manuscript needs a lot of work. First, the manuscript is too long. Second, the focus is on data, not what the use of the data should be for land use planning. The land use planning in the Southeast is famously weak and sprawl is a huge problem. The data appear to confirm this. Third, the time frame of 1945 to 2012 is OK, but there are too many graphics (such as 3(a) to 3(e) and studies with different time frames (Table 3) that result in a lot of confusion.
You need to provide overall results for the region: population change from 1950 to 2010 using census data; change in agricultural acres 1948-2012 from the agricultural census; and change in forest cover 1945 and in 2012. These are your three main drivers of land use change. You can then present the data for each state in 1945 and 2012. Then the graphics of land use change for each state will make more sense.
In Figure 4, you present 8 drivers of land use change without discussing them in any detail.
Finally, you really don’t deliver on your statement in the Abstract about “monitoring, managing, and prioritizing large area conservation practices.” There is a lot of preserved land in parts of the SEUS—especially Florida, South Carolina, North Carolina, and Virginia.
Specific items to address:
Define LULCC in the Introduction age 1 Line 42.
Page 2 L 44 You need to provide the figure for SEUS agricultural land cover, not the global figure.
Page 3 L 115 You need to say “$56 billion a year in output.”
Put the Lit Review as Section 2.2
Put the Analysis of major land types as Section 3
Results should be Section 4 and 4.1 Land-type data analysis
The population change in SEUS 1990 to 2010 is especially important as this was an era of rapid population growth in the entire US.
Delete the graphics that combine all of the states (3(a)-3(e)). Not needed and very cluttered and hard to read.
Table 4 should be removed, and the literature summarized as land use change ranges and time frames. The individual studies appear to be literally all over the map and across time. Very confusing.

Author Response
Reviewer 2
- This is a good topic and of interest to readers of Land. But the manuscript needs a lot of work. First, the manuscript is too long. Second, the focus is on data, not what the use of the data should be for land use planning. The land use planning in the Southeast is famously weak and sprawl is a huge problem. The data appear to confirm this. Third, the time frame of 1945 to 2012 is OK, but there are too many graphics (such as 3(a) to 3(e) and studies with different time frames (Table 3) that result in a lot of confusion.
Noted, thanks for your comments
The manuscript was reorganized and submitted
- You need to provide overall results for the region: population change from 1950 to 2010 using census data; change in agricultural acres 1948-2012 from the agricultural census; and change in forest cover 1945 and in 2012. These are your three main drivers of land use change. You can then present the data for each state in 1945 and 2012. Then the graphics of land use change for each state will make more sense.
There was an estimated average forest land change of 12% over the time period; forest land fluctuated from 1945 – 2012 according to the ERS dataset. Agricultural land cover for the period is estimated to be 20% according to agricultural census data 1940 - 2012 (https://agcensus.library.cornell.edu/ Accessed 11/27/2022), and the average population grew significantly from 1950 – 2012 at a rate of 2.59% annually according to U.S. Census in the region (https://www.census.gov/data/tables/time-series/dec/popchange-data-text.html Accessed 11/27/2022).
- In Figure 4, you present 8 drivers of land use change without discussing them in any detail.
The direct natural drivers (E.g., Hurricanes, floods, and fires) have a greater effect on landscape changes than indirect natural drivers (E.g., Topography, habitat, and invasive species). The outcome can lead to severe effects of the land cover in the region, effects on the regions climate and ultimately the land use in the future [7,100]. However, the artifical drivers (E.g., Population growth, agriculture, and industrialization) are deemed as anthropongenic. Their effects are at a greater rate and is rapidly carried out in the SEUS. This was also reported by Nepton et al. [11] relating to the drivers that affects LULCC. The direct artificial drivers (E.g., Land use planning and population growth) seem to have one of the highest impacts in the region. Our results on population growth show that the population continues to grow at 2.69%/year, with some states (Florida) having a higher growth margin than others [45]. This increase the need for land use planning, monitoring, and management in the region, which see more landscape being converted from one land type to another. The indirect artificial drivers (E.g., Policy and economic development) are also important. The decisions made around these drivers give rise to land use resolutions and the impacts goes beyond landscape changes [101]. These decisions affect the outcome of land use for other artificial drivers (E.g., Technology and agriculture) that have shown significant impact on the environment. Agriculture is identified closely to urbanization because its impacts affects the ecosystem [67], hydrology and habitat [41]
- Finally, you really don’t deliver on your statement in the Abstract about “monitoring, managing, and prioritizing large area conservation practices.” There is a lot of preserved land in parts of the SEUS—especially Florida, South Carolina, North Carolina, and Virginia.
Thanks for your comments
The manuscript was streamlined and the objectives changed, this particular section was deleted for it to be more specific to the objectives
- Specific items to address:
- Define LULCC in the Introduction age 1 Line 42.
Thanks for this comment, this was an oversight.
Land use and land cover change (LULCC)
- Page 2 L 44 You need to provide the figure for SEUS agricultural land cover, not the global figure.
According to the Economic Research Service (ERS) 2012 report, cropland in the SEUS account for approximately 14.4% (90.3 million acres) of cropland [13]
- Page 3 L 115 You need to say “$56 billion a year in output.”
Noted, this was corrected
- Put the Lit Review as Section 2.2
Noted, this was corrected
The flow of the manuscript was changed to make it more brief and the methodology clearer
- Put the Analysis of major land types as Section 3
Noted, this was corrected
- Results should be Section 4 and 4.1 Land-type data analysis
Thanks for your comments.
The necessary changes were made in the manuscript.
- The population change in SEUS 1990 to 2010 is especially important as this was an era of rapid population growth in the entire US.
Population changes were calculated using U.S. Census data from 1950- 2012, we stick to the specific time frame that we used for the manuscript
the average population grew significantly from 1950 – 2012 at a rate of 2.59% annually according to U.S. Census in the region (https://www.census.gov/data/tables/time-series/dec/popchange-data-text.html Accessed 11/27/2022).
- Delete the graphics that combine all of the states (3(a)-3(e)). Not needed and very cluttered and hard to read.
Thanks for your comments.
Graphs were deleted
Table 4 should be removed, and the literature summarized as land use change ranges and time frames. The individual studies appear to be literally all over the map and across time. Very confusing.
Noted, Thanks for your comments.
Table was removed along with the literature

Round 2
Reviewer 2 Report
The manuscript is improved, but several stylistic issues remain.
Specifically,
L 41 delete etc. and replace with forestry
L 120 $56 billion - add the $ - I mentioned this before.
L 458-464 delete - This discussion refers to projected outcomes in Florida, which are not covered by your study data.
L 466 observed Not declared
L 552 increases Not increase
L 601 has increased Not increases
L 613-614 land cover in the SEUS Not in protected areas - Your study does not identify protected areas in the SEUS.
Author Response
Reviewer 2 – Second round
The manuscript is improved, but several stylistic issues remain.
Specifically,
1. L 41 delete etc. and replace with forestry
Noted; thanks for your comments, this will be addressed
2. L 120 $56 billion - add the $ - I mentioned this before.
Noted; thanks for your comments, I apologize, this was an oversight.
3. L 458-464 delete - This discussion refers to projected outcomes in Florida, which are not covered by your study data.
Noted; thanks for your comments; this will be addressed
4. L 466 observed Not declared
Noted; thanks for your comments
5. L 552 increases Not increase
Noted; thanks for your comments
6. L 601 has increased Not increases
Noted; thanks for your comments
7. L 613-614 land cover in the SEUS Not in protected areas - Your study does not identify protected areas in the SEUS.
Noted; thanks for your comments
